# MuTr: Multi-Stage Transformer for Hand Pose Estimation from Full-Scene Depth Image

**DOI:** 10.3390/s23125509

**Published:** 2023-06-12

**Authors:** Jakub Kanis, Ivan Gruber, Zdeněk Krňoul, Matyáš Boháček, Jakub Straka, Marek Hrúz

**Affiliations:** 1Department of Cybernetics and New Technologies for the Information Society, University of West Bohemia Technická 8, 301 00 Pilsen, Czech Republic; zdkrnoul@kky.zcu.cz (Z.K.); matyas.bohacek@matsworld.io (M.B.); strakajk@kky.zcu.cz (J.S.); mhruz@ntis.zcu.cz (M.H.); 2Gymnasium of Johannes Kepler, Parléřova 2/118, 169 00 Prague, Czech Republic

**Keywords:** hand pose estimation, neural network, transformer, multi-stage

## Abstract

This work presents a novel transformer-based method for hand pose estimation—DePOTR. We test the DePOTR method on four benchmark datasets, where DePOTR outperforms other transformer-based methods while achieving results on par with other state-of-the-art methods. To further demonstrate the strength of DePOTR, we propose a novel multi-stage approach from full-scene depth image—MuTr. MuTr removes the necessity of having two different models in the hand pose estimation pipeline—one for hand localization and one for pose estimation—while maintaining promising results. To the best of our knowledge, this is the first successful attempt to use the same model architecture in standard and simultaneously in full-scene image setup while achieving competitive results in both of them. On the NYU dataset, DePOTR and MuTr reach precision equal to 7.85 mm and 8.71 mm, respectively.

## 1. Introduction

The task of 3D hand pose estimation has great potential in many real-world tasks such as virtual and augmented reality [1], robotics [2], medicine, automotives and sign language processing [3]. Moreover, hand pose estimation presents beneficial cues for action recognition [4,5]. In these tasks, it is crucial to know the pose of the hand correctly in a 3D scene. Hand pose estimation from a full-scene image is usually based on two main separate sub-tasks—hand localization and pose estimation—and some optional ones—input data representation, localization refinement, hand pose refinement, etc.—while each of these sub-tasks usually employs a different model.

Convolutional neural networks (CNNs) have dominated most computer vision tasks during the last few years [6,7,8]. They are used mainly for sub-tasks of hand localization and hand pose estimation [9]. Since the introduction of Vision Transformer [10], transformer-based approaches have become increasingly popular [11,12,13,14,15]. In principle with much lower inductive bias compared to CNNs, when enough data are available, transformers can surpass CNN performance by a significant margin.

The problem of estimating the 3D hand pose directly from the full-scene image via a single model has not yet been sufficiently explored. Inspired by the success of the transformers not only in classification tasks but also in detection and semantic segmentation tasks [16,17,18,19,20,21], we modified the well-know DETR approach [17] to suit the pose estimation task. Our modification of the model lies in the different use of the query vectors. We called this novel approach Deformable Pose Estimation Transformer (DePOTR) and achieved competitive results on standard hand pose benchmark datasets. To demonstrate the full potential of DePOTR, we propose a novel Multi-stage Transformer (MuTr) which completely omits the hand localization sub-task. While using this setup, we still achieve competitive results, although the input into the estimator is the full-scene image. Moreover, we show the ability to learn long-range dependencies between visual features to alleviate the computational complexity of the input data pre-processing. Our code is available at: https://github.com/mhruz/POTR (accessed on 16 May 2023).

Our main contributions can be summarized as follows:We propose a novel method for hand pose estimation based on transformer architecture—DePOTR—overcoming other state-of-the-art transformer-based methods and achieving comparable results with other non-transformer-based methods in the standard setup.We introduce a multi-stage approach—MuTr—which achieves competitive results while predicting 3D hand pose directly from the full-scene depth image via one model and replaces several separate sub-tasks in hand pose estimation pipeline by overcoming the tedious data processing.

## 2. Related Work

In this section, we review the related work on recent hand localization and pose estimation methods. We refer readers to [22,23] for an overview of previous methods, mainly [22] for data-driven and [23] for generative methods. The current neural networks for pose estimation are mainly based on convolutional and transformer architectures or their combination.

CNN-based hand pose estimation benefits from the projective mapping of input or output data and learning of 2D or 3D features [22,24,25,26,27]. In [28]; anchors evenly distributed in the input depth image are used to predict the offsets of the joints (A2J). There are two branches, one to handle the spatial offsets and another to handle the depth offset. Furthermore, anchor informativeness is predicted to suppress the influence of irrelevant anchors on the position of a given joint. Similarly, in [29,30], a differentiable re-parameterization module is designed to build spatial-aware representations from joint coordinates and use them in several stages. The predicted pose must be decoupled from the real-world coordinates as the additional post-processing step.

In contrast, in [31,32,33,34], pre-processing of the input data is needed to convert the scene image to the 3D point cloud. Hand PointNet [31] assumes point-wise estimations in heatmaps and unit vector fields defined on the input point cloud. It defines the closeness and the direction of every point in the point cloud to the hand joint. Two new methods based on the transformer structure have been introduced for hand pose and multiperson body pose estimation [34,35]. Recently, a key-point transformer has been proposed to estimate the pose of two hands in close interaction from a single color image [36]. The solution relies on 2D keypoint detection and self-attention module matching CNN features at these keypoints to the corresponding localization of hand joints.

Most of these works depend on input data that contain correctly detected images hand cropped from the full-scene image. Early works use simple depth map thresholding as a rough detection center of mass and a bounding box defined around them. Otherwise, most other works adapt the hand localization network proposed in [24] (V2V-PoseNet). Only a few works deal with hand pose estimation from the full-scene image as one complex problem [37,38,39]. In [37], the authors propose the 2D object detection method via three neural network models, and similarly to [38,39], localization networks are used in one pipeline solution. Following this research, we propose a new transformer architecture that can also be innovatively tailored for full-scene images as one model solution.

## 3. Methods

We introduce a novel Multi-stage Transformer method—MuTr—solving the problem of hand pose estimation from full-scene images of the whole body of a single user. For the stage steps, the new Deformable Pose Estimation Transformer (DePOTR) model is designed to address precise 3D estimation; see the scheme in Figure 1. The modalities of the training data can affect the final accuracy of a given predictor. To test the sensitivity of our method to it, we consider several data modalities.

### 3.1. Training Data Modalities

Most of the imaging techniques are based on the pixel coordinate system (u,v) as a result of perspective transformation P of real-world objects in a camera coordinate system (x,y,z), for 2.5D images $D_{im}$ of size (m×n) (depth maps) where $D_{im}\left(u,v\right)\in R$ captures *z* coordinate as the distances of the object from the image plane.

The original design of 3D hand pose estimation expects cropped and normalized images $D_{im\_crop}$ as the input. These images are centered on the hand and do not contain any unwanted outer areas of the full-scene image $D_{im}$ [9,40,41]. We can rewrite this as follows:(1)$$D_{im\_crop}=C_{norm}\left(A_{sT}\left(D_{im}\right)\right),$$
where $D_{im\_crop}$ is a cropping image, $A_{sT}$ is 2D similarity transformation performing scale and translation of the full-scene image $D_{im}$ according to a cubic box and $C_{norm}$ is a cropping operation with depth value normalization; see [41] for more details. The position and size of the cubic box must be given for each full-scene image separately.

Let $X_{3Djoint}$ be an N×3 vector of 3D labels of *N* hand joints. An inconsistency exists between $D_{im\_crop}$ and $X_{3Djoint}$ defined in the coordinate system (x,y,z). We solve the inconsistency by applying a 3D ↔ 2.5D transformation on the input or output training data.

**3D transformation of the input:** Following [42], we define the transformation of each pixel from the pixel coordinate system into the camera coordinate system by applying the intrinsic parameters of the depth camera as:(2)$$D_{im\_3Dcrop}=C_{norm}\left(T_{3Dproj}\left(D_{im}\right)\right).$$

We define a point cloud of 3D points $P_{i}=\left(x,y,z\right)$, i=1..M, where *M* is the number of all pixels in $D_{im}$. Next, our method discretizes $P_{i}=\left(x,y,z\right)$ onto the XY plane with the coordinates $\left(u^{\prime },v^{\prime }\right)$ and pixel value as $\min\left(P_{j}^{z}\right)\;\forall j\in \left(u^{\prime },v^{\prime }\right)$, where $u^{\prime }≐x$ and $v^{\prime }≐y$. In general, the transformation does not guarantee pixel value continuity in the new coordinate system and the standard interpolation technique could be considered to insert missing pixel values in $D_{im\_3Dcrop}$.

**2.5D transformation of the output:** As the second option, the inconsistency is solved by transforming the output data as:(3)$$X_{2.5Djoint}=T_{2.5Dproj}\left(X_{3Djoint}\right).$$

The predictor model estimates hand joints $X_{2.5Djoint}=\left(u,v,z\right)$ that must be back-projected to the camera coordinate system (x,y,z) via a post-processing step applying the intrinsic parameters of the depth camera. This option generalizes the original solution of [40] and follows the idea of coordinate decoupling in [30]. We further investigate the effect of these modalities in an experiment.

### 3.2. Deformable Pose Estimation Transformer

The proposed DePOTR model is based on the Deformable DETR model [17], which is an extension of the DETR model [16]. In the DETR pipeline, the input image is processed via a backbone CNN that produces a feature map. The map is transposed into a sequence of feature vectors with a 2D positional encoding. The sequence is inputted into a transformer, outputting a predetermined number of hypotheses with detected objects—their class and relative position in the image. The *no object* class is also considered and thus a variable number of objects can be detected. Deformable DETR differs in the computation of the self-attention in the encoder, where only a small subset of all possible pairs of features is considered. For each feature vector, the locations of the features attended to are learned by the model and are dependent on the query feature vector.

The deformable attention module uses the formula:(4)$$\begin{array}{cc} \mathrm{DeA}(z_{q},p_{q},x)= & \\ \sum_{m=1}^{M}W_{m}\sum_{k=1}^{K} & A_{mqk}\cdot {W^{\prime }}_{m}x\left(p_{q}+\Delta p_{mqk}\right), \end{array}$$
where x is the input feature map, *q* is the index of the query described by its feature vector $z_{q}$ (in the original DETR paper [16] it is referred to as learned positional embeddings—*object queries*) and 2D reference point $p_{q}$, *m* is the index of the attention head, *k* is the index of the sampled key, where *K* is much smaller than all the possible keys. $\Delta p_{mqk}$ and $A_{mqk}$ denote the sampling offset and attention weight of the $k^{th}$ sampling point in the $m^{th}$ attention head, respectively. $W_{m}$ and $W_{m}^{\prime }$ are matrices of learned linear layers. Lastly, the Deformable DETR model takes advantage of the different scales in individual layers of the backbone CNN. The deformable attention module is augmented so that it can access the different scales in the attention computation. The multi-scale deformable attention (MSDeA) module is defined as:(5)$$\begin{array}{c} \mathrm{MSDeA}(z_{q},{\hat{p}}_{q},{\left\{x^{l}\right\}}_{l=1}^{L})=\\ \sum_{m=1}^{M}W_{m}\sum_{l=1}^{L}\sum_{k=1}^{K}A_{mlqk}\cdot {W^{\prime }}_{m}x^{l}\left(\phi_{l}\left({\hat{p}}_{q}\right)+\Delta p_{mlqk}\right). \end{array}$$

${\left\{x^{l}\right\}}_{l=1}^{L}$ represents the multi-scale feature maps; each feature map $x^{l}$ can have arbitrary size. ${\hat{p}}_{q}\in {\left[0,1\right]}^{2}$ is the relative coordinate of the query in each feature map. The function $\phi_{l}$ maps the relative position of the query into the absolute position in the relevant feature map. The attention weights $A_{mq}$ sum up to one. A positional embedding is added to the input vectors of the encoder. Deformable DETR adds an additional positional embedding representing the layer that the input feature $x^{l}$ comes from.

Our modification of the model lies in the different use of the query vectors. We define a precise number of query vectors equal to the number of joints that we are detecting. Each query vector is decoded into a 3D location only, the class (joint identity) is omitted. The identity of the joint is given simply by its index *q* and is constant during the training. This must also hold for the target data (i.e., each joint has its own unique index *q*). These modifications change the form of the loss function since we are interested only in the location error. The form of the transformer that uses a predetermined number of queries is referred to as a non-autoregressive transformer [34].

**Training:** The input of our DePOTR model is a depth image containing a hand. Since we use a pre-trained backbone model trained on RGB data, we copy the single channel of the depth image into three channels. The backbone CNN processes the input and computes the individual feature maps ${\left\{x^{l}\right\}}_{l=1}^{L}$. These are then processed by the transformer model which outputs a predetermined number of 3D joint locations. These are compared with the targets via the smooth L1 loss function:(6)$$L=\frac{1}{N_{j}}\sum_{i}^{N_{j}}\sum_{j}^{3}L1_{smooth}\left(y_{j}^{i},{\hat{y}}_{j}^{i}\right),$$
(7)$$L1_{smooth}\left(x,y\right)=\left\{\begin{array}{cc} 0.5\cdot {\left(x-y\right)}^{2}, & \mathrm{if}\;|x-y|<1\\ |x-y|-0.5, & \mathrm{otherwise}, \end{array}\right.$$
where $y_{j}^{i}$ and ${\hat{y}}_{j}^{i}$ is the *j*th component of the *i*th target joint and the predicted joint, respectively, and $N_{j}$ is the total number of joints. This loss is back-propagated through the model in an end-to-end fashion.

**Augmentations:** We use several standard augmentations—in-plane rotation, scale and translation. Furthermore, we apply several less common augmentations. To suppress the effect of common depth sensors that produce perforated images of small objects such as hands, we apply a gray-scale morphological close operation with a fixed kernel size to every training sample. We use this augmentation during prediction also. In Table 1, this augmentation is dubbed *prep.* (i.e., preprocessing). Another uncommon augmentation is re-scaling (*rescal.* in Table 1). This augmentation first re-samples the input into a lower resolution and then up-scales it back to the original resolution using nearest-neighbor interpolation. This operation simulates the appearance of hands that are far away from the camera and are thus projected into a lower number of pixels. To simulate different shapes of hands, we use gray-scale morphological erosion and dilation (*E&D aug.*). Lastly, we use crop-out augmentation, which randomly replaces the cropped-out region with either the foreground or the background value. To our knowledge, this augmentation is not used in prior hand pose estimation works. We apply the augmentations on-the-fly with a defined probability.

### 3.3. Multi-Stage Transformer

The common approaches for hand pose estimation are based on two separate sub-tasks. In the first sub-task, a hand is detected in a full-scene image. The second sub-task finds the positions of the hand joints themselves. To show the full potential of our DePOTR approach, we decided not to solve the first sub-task separately, but as the first stage of a multi-stage “end-to-end” solution—MuTr (Multi-stage Transformer). That is, in the first stage, an estimate of the hand joint pose is made from the full-scene image. In the next stage, this estimate is used to crop and modify the original input image. From the predicted joint positions, the center of mass (CoM) is computed as the mean value ${\bar{X^{\prime }}}_{C}=\left(u_{C},v_{C},z_{C}\right)$ and then used together with the minimum and maximum values ($u_{min},u_{max}$), ($v_{min},v_{max}$) of estimated joint coordinates to modify the input image to the cube box that ideally contains only the hand. Values of all image pixels outside of the ranges $[u_{C}-\mathrm{range}\_\mathrm{half}$ (125 px for NYU and 175 px for HANDS2019 Task 1 datasets), $u_{C}+\mathrm{range}\_\mathrm{half}],\;\left[v_{C}-\mathrm{range}\_\mathrm{half},v_{C}+\mathrm{range}\_\mathrm{half}\right],\;\left[z_{C}-125\;\mathrm{mm},z_{C}+125\;\mathrm{mm}\right]$ are replaced by the background value and the resulting depths in the cropped image are thus centered according to the value $z_{C}$ and normalized to [−1, 1]. An image of size of $\left[u_{min}-25\;\mathrm{px},u_{max}+25\;\mathrm{px},v_{min}-25\;\mathrm{px},v_{max}+25\;\mathrm{px}\right]$ is cropped. This image is then resized to the desired size (224 × 224 pixels in our case) and used as input for the next stage of the estimation.

The process of improving the joint pose estimate by refining the region of interest in the input image can generally be repeated *n* times. In our ablation study, we investigated up to the four-stage estimation of the hand pose.

## 4. Experiments

We evaluate DePOTR and MuTr on four publicly available depth data datasets designed for hand pose estimation tasks: NYU [40], ICVL [43], HANDS 2017 [44] and HANDS 2019 Task 1 [45].

### 4.1. Datasets and Evaluation

NYU contains 72,757 training and 8252 testing RGB-D images and 3D annotation of the hand pose. The training set contains one person only while the test set has two persons in total (the first one is the same as in the training set). For the training, we used depth data from the frontal view only. According to the standard dataset protocol, we used 14 out of 36 joints during the evaluation step. To make a fair comparison of our results with prior works, we used the same practice of using two different sizes of cube boxes for the two different performers (300 mm and 250 mm, respectively).

The ICVL dataset contains 22k training real images and 1.6k test real images. The dataset also contains an additional 300k augmented images with in-plane rotations which we also used for the training. The dataset captures ten different subjects with 16 annotated hand joints. It is a relatively small dataset, but we decided to include it in our work because it is the dataset where the previous work using transformers achieves the strongest results.

The HANDS2017 dataset is one of the largest benchmark datasets for 3D hand pose estimation. It consists of 957k training and 295k test RGB-D images sampled from the BigHand2.2M [46] and F-PHAB datasets [4]—10 people with 21 annotated hand joints. The training set contains five people, while the test set contains ten people (five original ones and five unseen people). The hand pose is defined as 3D positions of 21 joints (16 joint positions and 5 fingertip positions). To evaluate the results, we used the Codalab competition system (https://competitions.codalab.org/competitions/17356, accessed on 16 May 2023).

The HANDS2019 Task 1 dataset contains 175k training and 125k test images and is again based on the BigHand2.2M [46] dataset with an emphasis on containing the challenging data samples.

We evaluate DePOTR and MuTr using two commonly used metrics. The first metric is the average 3D joint error computed over all test images. The error is computed as the Euclidean distance between the predicted and ground-truth positions in mm. The second metric is the proportion of good estimates out of all test images at a given accuracy level. An estimate is considered good only if the error of each joint is below a given threshold.

### 4.2. Implementation Details

For evaluation purposes, the localization network proposed in [24] is used to obtain an accurate bounding box and crop the image. The input data modalities are normalized to the range [−1, 1] according to the constant size of the cubic box with an edge size of 250 mm (the only exception already mentioned above is the NYU dataset). We use a standard Deformable DETR set of training parameters in all experiments unless otherwise noted. This includes AdamW optimizer with starting learning rate (*lr*) and backbone learning rate (*blr*) of 1 × 10^−5^ with *lr* drop after 40 epochs, batch size of 10 (16 for evaluation for direct comparison with [34]), weight decay of 1 × 10^−4^, gradient clipping of 0.1, *ResNet50* as the backbone, sine positional encoding, 6 encoding and 6 decoding layers, dimension of feedforward layers in the transformer block of 1024, size of embedding of 256, dropout of 0.1 and 8 attention heads. We used the following values for augmentations: relative shift [−0.1, 0.1], scale [0.8, 1.2], rotation [−40°, 40°], erode [3, 5] pixels, dilate [3, 7] pixels, re-scaling [0.5, 0.9]. All experiments were performed on a single NVIDIA GeForce 1080 Ti GPU using the PyTorch framework [47]. We consider this training setup to be the baseline setup.

### 4.3. Ablation Study

The ablation study is conducted on the NYU dataset because it has a rich range of captured hand poses and its training set is limited to a single subject, while its evaluation part includes an additional unseen speaker.

**DePOTR:** We consider both training data modalities: *3Dproj* (Equation 2) and *2.5Dproj* (Equation 3). We compare these modalities with the *Baseline* pipeline (Equation 1) which does not include any optional sub-tasks, i.e., input data representation or localization refinement.

A summary of the ablation study can be found in Table 1. Inconsistency correction in the output data while using *2.5Dproj* improves the results over the *Baseline* from 11.11 mm to 10.37 mm. On the other hand, *3Dproj* only slightly improves the results to 10.25 mm while being much more computationally expensive when compared with *2.5Dproj*. This experimental result indicates that transformers are capable of learning their internal 3D projection, so the use of *3Dproj* for full-scene images is unnecessary.

In the next step, we apply preprocessing to the input data, and erode and dilate augmentations and re-scaling. This further improves the results to 9.91 mm. We used an extended training scheme (160 epochs, *lr* and *blr* 1 × 10^−4^, *Resnext50_32x4d* [48] backbone—*enhance*) and added crop-out augmentation and obtained the result 8.62 mm. Finally, we achieved our best result 7.85 mm using more sophisticated backbone *EfficientNet2_rw_m* [49] (implementation taken from Pytorch Image Models (https://github.com/rwightman/pytorch-image-models, accessed on 16 May 2023)) with rotation augmentation values extended to [−180°, 180°]—*enhance2*.

**MuTr—Full-Scene Image:** A summary of the ablation study can be found in Table 2. In the first stage, the input full-scene image is resized to 224 × 224 pixels and then normalized by the difference between the highest and the lowest depth values of the training dataset. The baseline DePOTR parameter settings and *2.5Dproj* input modality are used. We refer to the model with this training setup as *S1 baseline*. It can be seen in Table 2 that by using a better backbone network and a modified training scheme enhance2 (but only with 100 training epochs), a significant improvement of the results is achieved (31.92 >> 21.57)—*S1 enhance2* (20.24—*S1${}_{ct}$ enhance2* (using the “cube trick” with the different depth data cube sizes—default 250 and smaller 208 mm (250 × 250/300))).

Designation of the second stages is based on the corresponding first stages and their training setups (i.e., *S2 baseline* uses the joint estimates from the *S1 baseline* and the same training scheme, etc.). The *S2 S1 init* is exactly the same as the *S2 baseline*, except that the best-trained weights from the previous phase of the *S1 baseline* were used to initialize this stage. The *S2 enhance2* uses the output of the *S1 enhance2* and the same training scheme without initializing the weights (because the previous *S2 S1 init* experiment showed that this is not beneficial at this stage). Again, a significant improvement in the results can be observed (21.57 >> 10.44)—*S2 enhance2* (9.11—*S2${}_{ct}$ enhance2*).

The third stage *S3 baseline* builds on *S2 baseline*. *S3 S2 init* again uses the weight initialization from the previous stage. This leads to a significant improvement in the result (15.49 > 14.41). *S3 enhance2* using the output of *S2 enhance2* is then our best overall result of 9.84 mm (8.71—*S3${}_{ct}$ enhance2*). The fourth stage brings no improvement—*S4 enhance2*. Since, according to [50], the poses with an error lower than 20 mm approach the limit of human accuracy for close-by hands, even the result of the first stage can be used in this case for some tasks.

### 4.4. Comparison with State-of-the-Art

From the mean prediction error results in Table 3, it can be seen that our approach outperforms the previous work H-trans, which also uses the transformer architecture in both accuracy and estimation speed. Furthermore, DePOTR significantly outperforms the results of V2V-PoseNet and the A2J approach. DePOTR is only slightly outperformed by the SRN and SSRN solutions only. In the case of the maximum prediction error, Figure 2, it can be seen that our solution outperforms all solutions between the 20 and 30 mm threshold. Unfortunately, H-trans data for this metric are unavailable, so we can only make an indirect comparison. From the plot in Figure 3 presented in the paper [34], H-trans is inferior to the V2V-PoseNet method up to a threshold of about 25 mm. Nevertheless, DePOTR outperforms V2V-PoseNet over the entire range of values.

For the ICVL dataset—see Table 3—DePOTR achieves the best result. The graph in Figure 2 shows that DePOTR outperforms the other methods. Once again, we do not have the curve data for the H-trans method, but from the indirect comparison via Figure 3 from [34], H-trans is outperformed by the V2V-PoseNet method once again until a threshold of about 18 mm, whereas DePOTR outperforms V2V-PoseNet over the entire range of values. In the case of the HANDS2017 dataset (Table 4), we report our results using the backbone EfficientNet2_rw_m, 25 epochs and *lr* 1 ×$10^{-4}$ and *blr* 1 ×$10^{-5}$ with drop after 10 epochs. We chose the epoch with the best performance on the portion of 29.5k test images but report the results for all 295k test images. We outperform the overall result of V2V-PoseNet and A2J in the case of seen data. The SSRN method achieves the best results overall.

With our three-stage solution, we were even able to outperform our standard DePOTR solution ${\mathrm{Ours}}_{c300}DePOTR$; see Table 2—9.84 mm (8.71) vs. 10.83 mm (7.85) (where c300 means using of a cube of the size of 300 mm for all test data (i.e., no cheating with cube sizes as in the case of the common use of two cube sizes of 300 and 250 mm, respectively)). From the graph in Figure 2, it can be seen that our MuTr solution (Ours_staged2, which uses the “cube trick”) overcomes both V2V-PoseNet and A2J solutions from a threshold of about 18 mm. In the case of the HANDS2019 Task 1 dataset (Table 5, 90 epochs, *lr* 1 ×$10^{-4}$ and *blr* 1 ×$10^{-5}$), we achieved the respectable score of **15.64** mm which is on par with the score of 15.57 mm of the V2V-PoseNet method (NTIS solution—deeper architecture than the original with averaging of N-best predictions from six different epochs) published in [45] and which is only about less than 2 mm behind the best solution of 13.66 mm from that work. In Table 3, in the part headed “Full-Scene Image”, we can see that our MuTr solution of 8.71 mm outperforms all other previous full-scene image solutions by a wide margin.

To further compare MuTr with existing approaches, we modify the SRN architecture to a multi-stage full-image solution (see SRN-FI in Table 3). We changed the input data size to 256 × 256 and the cube size to 750 × 750 × 750 mm, now placed constantly in the center of the full-scene image. This resolution leads to a pool factor of 8, where an additional 2D average pooling of 4 × 4 with stride 2 adjusts the size of the extracted feature maps. Data augmentation includes random scaling ([0.85, 1.15]) and translation ([−5, 5]). It can be seen that the first stage of the MuTr solution outperforms the SRN-FI with default two-stage architecture—20.24 mm vs. 39.97 mm.

## 5. Attention Analysis

To better understand the decision process of both of our models, we analyze “what they are looking at” during the decision process. There are two main interesting internal interpretations to consider with respect to this problem—self-attention and cross-attention.

Figure 3 displays average DePOTR self-attention for all 14 hand joints. It can be seen that in most cases each joint has the highest self-attention to itself, with few interesting exceptions. First, for the prediction of the position of the center of the hand, the positions of the fingertips are particularly important. We believe this is quite intuitive behavior, because fingertips are arguably easier to distinguish from the other image parts than the hand center, while they also carry information about the approximate position of the hand center. Second, the second finger joints sometimes highly attend to the fingertip of the same finger. Once again, we believe the reason for this behavior is very similar to the behavior in the first exception. Looking at the other self-attention values, it is obvious that for predictions of finger joints, other joints of the same finger are very important. Moreover, sometimes joints of close fingers are also quite important.

Self-attention in the first stage of the MuTr model can be seen in Figure 4. In this case, self-attention is much more spread among all joints. We argue this is caused by the lower resolution of the hand itself due to the fact that the input to the first stage of MuTr is the whole un-cropped image. This fact stems from the image of the hand being less detailed and therefore the model cannot rely only on a single location in the image. Other than that, we observe behavior similar to the standard DePOTR model, i.e., high attention to fingertips during the prediction of hand center, high attention to itself during the prediction of fingertips, etc. The behavior of the MuTr model in the subsequent stages (i.e., stage 2 and further) is the same as the behavior of the standard DePOTR model.

Figure 5 depicts DePOTR cross-attention for different feature level sizes. Similarly, as in the original Deformable DETR, we expected that to correctly predict joint positions the model will look at the surroundings of the joint itself, i.e., cross-attention will correspond to the part of the image with the predicted joint. However, closer analysis revealed that this is not true for our model. Rather, DePOTR spreads its cross-attention over the parts of the image where it is probable it will find some hand parts. The model will find these interesting image parts during its training and subsequently predict the results from the surroundings of these learned positions—anchors. We observe small movements of anchors for different hand shapes and different hand positions. In other words, for example, if the same image will be augmented to move the hand left, all anchors will move left as well; however, their relative position is almost static. In our opinion, this counter-intuitive behavior is the result of the task change (object detection for the original DETR vs. pose estimation for DePOTR) and the positions of the anchors are dependent on the training set. This phenomenon definitely needs additional research and we are planning to focus on it in the future.

MuTr cross-attention (the first stage) can be seen in Figure 6. The behavior of the cross-attention is very similar to DePOTR cross-attention with the difference that in this case the anchors are concentrated near the middle of the image, which can be expected because the hand is usually located around the middle of the input image. MuTr cross-attention in the following stages has once again the same behavior as the cross-attention of the original DePOTR approach.

## 6. Conclusions

In this paper, we present DePOTR, a transformer-based method for hand pose estimation. DePOTR is successfully tested in two different pipelines. In the first pipeline—the standard one—the input into DePOTR is a region of interest with a detected hand. In this setup, our method overcomes other transformer-based methods while performing on par with other state-of-the-art approaches. In the second pipeline—MuTr—DePOTR estimates joint positions from the full-scene image in a multi-stage manner. This eliminates the need of designing a designated hand region detection algorithm. We experimentally show not only that DePOTR can be used without changes to architecture in the multi-stage pipeline, but also that it overcomes all the other tested multi-stage solutions.

Additionally, we provide extensive ablation studies for both presented setups with a comprehensive discussion about the functioning of the model with respect to its attention. Ablation study of different training modalities shows that the model using 2.5D data achieves results comparable to the one using 3D.

In our future research, we would like to focus on additional improvements to the MuTr approach. Specifically, we believe that a higher resolution of the input can improve results because the multi-stage pipeline is especially prone to loss of detail.

## Figures and Tables

**Figure 1 sensors-23-05509-f001:**
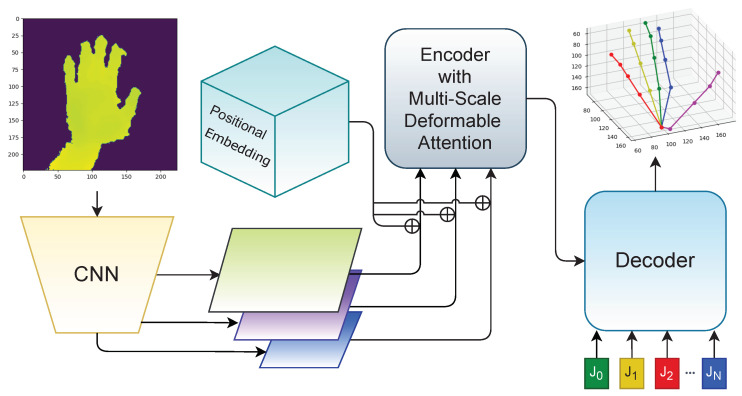
The scheme of our proposed DePOTR model.

**Figure 2 sensors-23-05509-f002:**
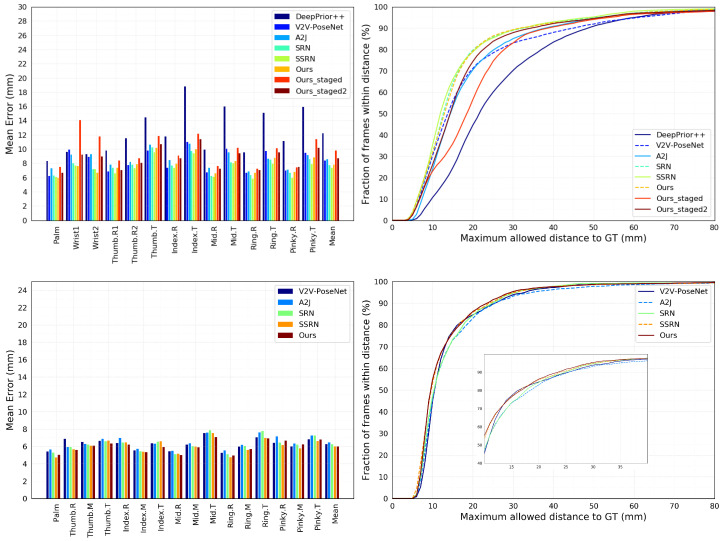
Comparison with state-of-the-art methods on NYU (upper line) and ICVL (lower line) datasets. Ours—DePOTR; Ours_staged—MuTr S3 enhance2; Ours_staged2—MuTr S3${}_{ct}$ enhance2 with the “cube trick”.

**Figure 3 sensors-23-05509-f003:**
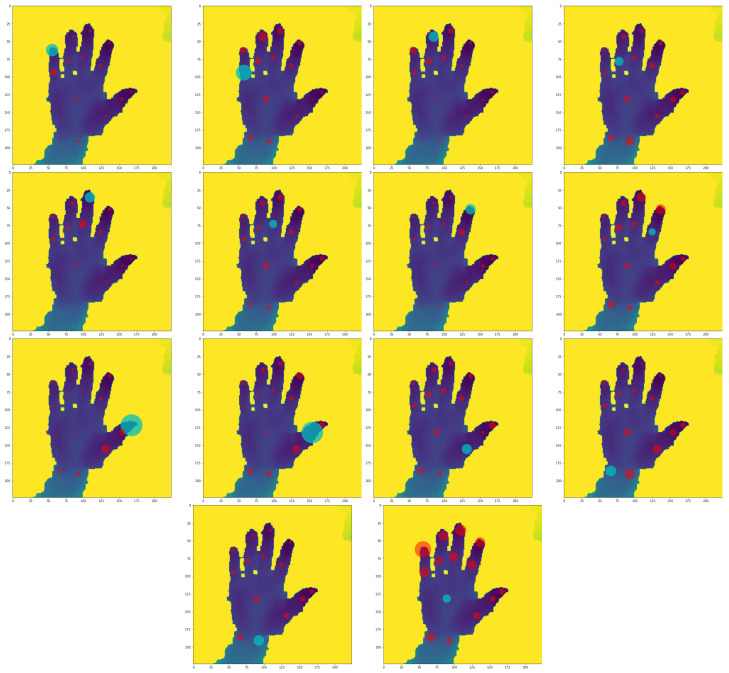
Average DePOTR self-attention for all 14 joints counted from 1000 predictions of randomly selected NYU test set images; depicted in the test image #133.

**Figure 4 sensors-23-05509-f004:**
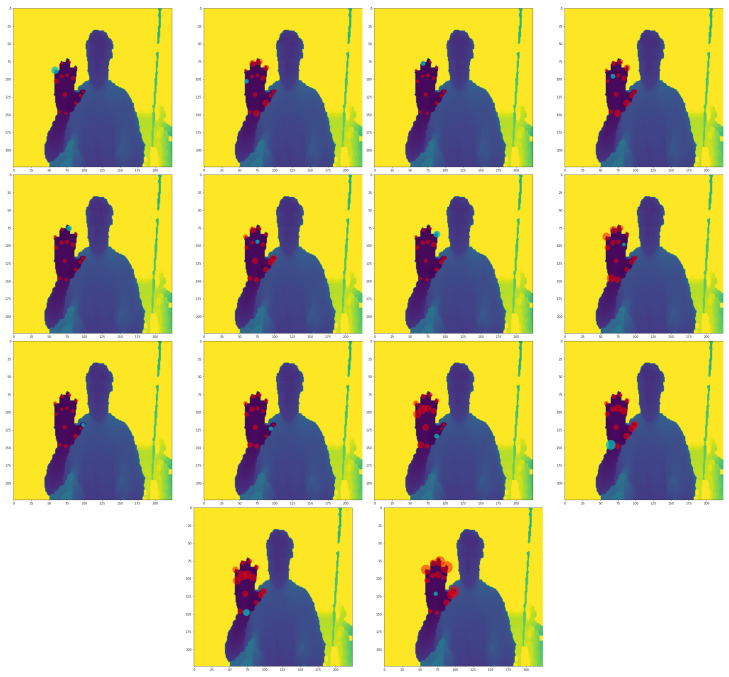
Average MuTr (stage 1) self-attention for all 14 joints counted from 1000 predictions of randomly selected NYU test set images, depicted in the test image #133.

**Figure 5 sensors-23-05509-f005:**
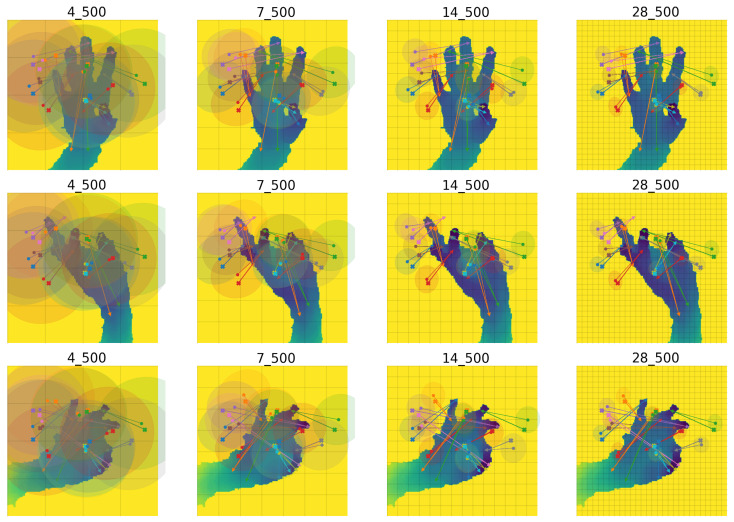
Examples of distribution of all DePOTR cross-attention points (500) for all 14 joints and for three different random test images; from the left to the right feature level sizes: 4, 7, 14 and 28. The same color cross point denotes the reference point, the bold point is the mean and the circle is the variance of all offset sampling points for one joint.

**Figure 6 sensors-23-05509-f006:**
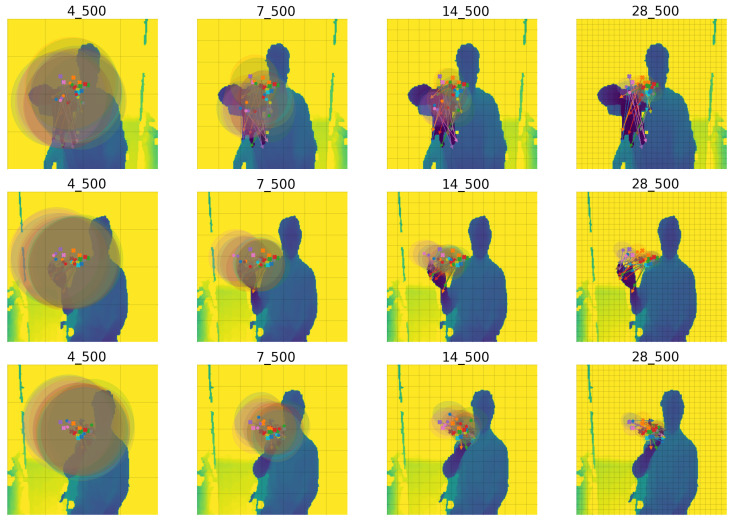
The examples of distribution of all MuTr (stage 1) cross-attention points (500) for all 14 joints and for three different random test images; from the left to the right feature level sizes: 4, 7, 14 and 28. The same color cross point denotes the reference point, the bold point is the mean and the circle is the variance of all offset sampling points for one joint.

**Table 1 sensors-23-05509-t001:** NYU DePOTR ablation study.

Method	Precision (mm)
Baseline	11.11
with 2.5Dproj	10.37
with 3Dproj	10.25
+prep. + E&D aug. + re-scal.	9.91
+enhance + crop-out aug.	8.62
+enhance2	**7.85**

**Table 2 sensors-23-05509-t002:** NYU MuTr ablation study.

Method	Precision (mm)
S1 baseline	31.92
S1 enhance	24.43
S1 enhance2	**21.57**
S1${}_{ct}$ enhance2	**20.24**
S2 baseline	15.69
S2 S1 init	15.92
S2 enhance	11.17
S2 enhance2	**10.44**
S2${}_{ct}$ enhance2	**9.11**
S3 baseline	15.49
S3 S2 init	14.41
S3 enhance	10.66
S3 enhance2	**9.84**
S3${}_{ct}$ enhance2	**8.71**
S4 enhance2	9.86
Ours${}_{c300}$ DePOTR	10.83
Ours DePOTR	7.85

**Table 3 sensors-23-05509-t003:** SOTA comparison—Precision on NYU and ICVL datasets and inference speed.

Method	NYU	ICVL	FPS
Deepprior++ [9]	12.24	8.10	30.0
A2J [28]	8.61	6.46	105.1
V2V-PoseNet [24]	8.42	6.28	3.5
SRN [29]	7.79	6.27	263.4
SSRN [30]	**7.37**	6.01	**295.6**
H-trans [34]	9.80	6.47	43.2
Ours DePOTR	7.85	**5.98**	106.9
Full-Scene Image			
WR-OCNN [39]	15.62	—	—
SRN-FI	39.97	—	—
Ours MuTr (only S1)	20.24	—	—
Ours MuTr	**8.71**	—	—

**Table 4 sensors-23-05509-t004:** HANDS2017 SOTA comparison.

Method	Avg. (mm)	Seen (mm)	Unseen (mm)
V2V-PoseNet	9.95	6.98	12.43
A2J	8.57	6.92	9.95
SRN	8.39	6.06	10.33
SSRN	7.88	5.65	9.75
Ours DePOTR	8.88	6.36	10.98

**Table 5 sensors-23-05509-t005:** HANDS2019 Task1 MuTr results.

Method	Precision (mm)
S1 enhance2	23.97
S2${}_{res}$ enhance2	**15.64**
S3${}_{res}$ enhance2	15.66

## Data Availability

NYU dataset https://jonathantompson.github.io/NYU_Hand_Pose_Dataset.htm; ICVL dataset https://labicvl.github.io/hand.html; HANDS2017 dataset http://icvl.ee.ic.ac.uk/hands17/challenge/; HANDS2019 dataset https://sites.google.com/view/hands2019/challenge (accessed on 16 May 2023).

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
