# Peer review of "MuTr: Multi-Stage Transformer for Hand Pose Estimation from Full-Scene Depth Image"

_sensors, 2023, doi:10.3390/s23125509_

Round 1
Reviewer 1 Report
This paper presentes a novel method for hand pose estimation based on transformer architecture.
Abstract:
1- The authors should present more specific details about
the methodology employed in DePOTR and MuTr.
2- The authors should include performance metrics and numerical results to support the claims.
3- The authors should the novel contributions of DePOTR and MuTr to the field of hand pose estimation.
Conclusion:
1- The authors should list briefly the key findings from the standard and MuTr pipelines.
2- The authors should explain the practical implications and potential applications of DePOTR.
3- The authors should provide address the impact of ablation studies.
4- The authors should provide future directions.
--
1- The authors used a point after (1). However, the sentence continues with where.
2- What is MSD? The authors should explain it.
3- The authors should discuss the implications of the observed behavior
and cross-attentions for the overall performance of DePOTR and MuTr.
Author Response
Dear reviewer,
Thank you very much for your comments and valuable insight into our manuscript. We will address each of them one by one.
Comments about Abstract
C: The authors should present more specific details about the methodology employed in DePOTR and MuTr.
R: We are not sure what specifically have reviewer in their mind. We mention that both models are based on transformer architecture directly in the abstract.
C: The authors should include performance metrics and numerical results to support the claims.
R: We added results to the abstract.
C: The authors should the novel contributions of DePOTR and MuTr to the field of hand pose estimation.
R: We believe our contribution is concluded by sentence: “MuTr removes the necessity of having two different models in the hand pose estimation pipeline.”
Comments about Conclusion
R: We significantly extended conclusion section to address all reviewer’s comments.
Other
C: The authors used a point after (1). However, the sentence continues with where.
R: We correct the mistake.
C: What is MSD? The authors should explain it.
R: We added an explanation of abbreviation to the text.
C: The authors should discuss the implications of the observed behavior and cross-attentions for the overall performance of DePOTR and MuTr.
R: Unfortunately, without additional experiments, we are unable to discuss this topic. We believe that the phenomenon occurring in cross-validation of the model is indeed interesting and unexpected. Despite it, the models reached very promising results.
Reviewer 2 Report
The manuscript introduces a model for end-to-end hand pose estimation based on the transformer paradigm.
The method performance is shown to be superior to that of other transformer-based solutions, and not superior (but on par) with non-transformer-based solutions.
The paper is well structured.
The presentation requires a revision, as some crucial concepts miss contextualization and definitions. The English writing presents some issues as well.
The analysis on attention is interesting and valueable.
The title must be changed to explicitly state that the input is a *depth* image, as declared later on in line 148.
A simple proposed change is "MuTr: Multi-stage Transformer for Hand Pose Estimation from Full-Scene Depth Image".
The abstract and introduction should also clearly mention this detail.
Section 3 mentions "Deformable POse estimation TRanformer - DePOTR model is designed to address precise 3D estimation and solve data modality inconsistencies between the input depth map and 3D pose labels".
The property of being designed to address data modality inconsistencies must be further elaborated.
First of all, it is not clear where and how this is addressed (see next point on Section 3.1).
Secondly, the efficacy is not clearly proven. Table 1 reports some experiments that are potentially related to this, but with little to no explanation.
Section 3.1 apparently describes the issue of determining correspondences between 2D, 2.5D and 3D data.
The goal of this section is not immediately clear.
For example, the authors should clarify whether the described transformations (equation 2 and 3) are part of the proposed solution, and how.
The introduction of equation 1 itself is not clear: " We can rewrite such linear transformation of the input image D_im". The sentece references a linear transformation that is never actually mentioned earlier. Please rephrase.
It would also help to define the cardinality of the involved variables. e.g. D_im € R^3 (for example).
X_joint in equation 3 is also not defined.
Section 3.2 explains that DePOTR is based on the existing DETR model.
This relationship should be made explicit from the very beginning (introduction section), with a synthetic description of the contribution (as reported in line 140 to 146).
The English writing presents some issues.
Some example sentences that are not clear or not well-formed.
Line 70:
"Most of these works depend on input data captured only correctly detected hand and cropped from the full-scene image". (?)
Line 96:
" It can be solved by applying a 2.5D ↔ 3D transformation applied on the input or output data" (applying applied)
Line 129:
" Lastly, the Deformable DETR model makes advantage of the different scales in individual layers of the backbone CNN." (takes advantage)
Line 157:
"We use several standard augmentations - in-plain rotation, scale," (should be "in-plane")
Author Response
Dear reviewer,
Thank you very much for your comments and valuable insight into our manuscript. We will address each of them one by one.
Comments:
The title must be changed to explicitly state that the input is a *depth* image, as declared later on in line 148.
A simple proposed change is "MuTr: Multi-stage Transformer for Hand Pose Estimation from Full-Scene Depth Image".
The abstract and introduction should also clearly mention this detail.
R: We addressed this problem and added word “depth” to the title of our paper. Moreover, we added it into the abstract and introduction as well.
C: The property of being designed to address data modality inconsistencies must be further elaborated.
R: We added explanation into Section 3 to make our statement clearer.
C: First of all, it is not clear where and how this is addressed (see next point on Section 3.1). Secondly, the efficacy is not clearly proven. Table 1 reports some experiments that are potentially related to this, but with little to no explanation.
R: We believe that we address this in the section Ablation study (to be more specific, line 253 and further). To make it clearer, we add additional explanation to this section.
C: Section 3.1 apparently describes the issue of determining correspondences between 2D, 2.5D and 3D data. The goal of this section is not immediately clear.
R: We extended the beginning of the Section 3.
C: The introduction of equation 1 itself is not clear: " We can rewrite such linear transformation of the input image D_im". The sentece references a linear transformation that is never actually mentioned earlier. Please rephrase.
R: We rewrite this section to make it easier to follow.
C: It would also help to define the cardinality of the involved variables. e.g. D_im € R^3 (for example).
R: We address this problem in the text.
C: X_joint in equation 3 is also not defined.
R: We defined X_joint in the text.
C: Section 3.2 explains that DePOTR is based on the existing DETR model. This relationship should be made explicit from the very beginning (introduction section), with a synthetic description of the contribution (as reported in line 140 to 146).
R: We added a new sentence to the introduction to make this relationship explicit.
C: The English writing presents some issues.
R: We addressed all the issues the reviewer mentioned.
Reviewer 3 Report
-----------------------
Summary
-----------------------
The paper « Multi-stage Transformer for Hand Pose Estimation from Full-Scene Image » proposes a transformer-based method for hand pose estimation.
Two pipelines are presented with firstly a ROI detection and secondly a joint position estimation.
This end-to-end model do not necessitate the classical 2-steps process with first the hand localization and second the pose estimation.
The experiments show convincing results on 4 datasets.
-----------------------
Comments
-----------------------
The paper is well written, well organized and easy to understand.
The study context is clearly introduced.
The related works are adequate.
The proposed method is deeply described and the detail experiments offer the reproducibility.
The Section 5 on attention analysis is particularly interesting with deep analysis on self- and cross-attentions.
-----------------------
Minor Concerns
-----------------------
Figure 2 (right part) could be improved with a zoom to 18 mm, where the proposed method outperforms V2V-PoseNet and A2J solutions.
Figure 2 (left and right parts) should use the same colors for the same methods between the upper and lower lines.
Table 5 should present the best results in bold and comments could be included about SSRN results over the proposed method.
Author Response
C: Figure 2 (right part) could be improved with a zoom to 18 mm, where the proposed method outperforms V2V-PoseNet and A2J solutions.
R: We added zoom.
C: Figure 2 (left and right parts) should use the same colors for the same methods between the upper and lower lines.
R: Unfortunately, we were unable to force our plotting tool to provide desired results. It assigns the colors automatically depending on the number of methods.
C: Table 5 should present the best results in bold and comments could be included about SSRN results over the proposed method.
R: We bolded the best result and added comment about SSRN to the text.
Round 2
Reviewer 2 Report
The paper can be accepted under the condition that the title is modified to explictly mention the input "depth" data.
The authors mentioned that they performed this modification in their response, but this is not the case.
A simple proposal is "MuTr: Multi-stage Transformer for Hand Pose Estimation from Full-Scene Depth Image".
Author Response
Dear reviewer,
We are truly sorry for the inconvenience. By mistake, we changed only \TitleCitation and not \Title in our tex file. It is corrected now.